# Magnitude of metabolic syndrome in Gondar town, Northwest Ethiopia: A community-based cross-sectional study

Solomon Mekonnen Abebe[1]*, Abayneh Girma Demisse[2], Shitaye Alemu[2], Bewketu Abebe[2], Nebiyu Mesfin[2]

1 Institute of Public Health, College of Medicine and Health Sciences, University of Gondar, Gondar, Ethiopia,
2 School of Medicine, College of Medicine and Health Sciences, University of Gondar, Gondar, Ethiopia

* solomekonnen@yahoo.com

**Data Availability Statement:** All relevant data are within the paper and its Supporting information files.

## Abstract

### Background

Metabolic syndrome (MetS) is becoming a major public health problem globally; it is clear that the burden of MetS is rapidly increasing the rates of non-communicable diseases (NCD). In Ethiopia studies done so far have shown a large disparity in magnitude of the prevalence of MetS and were mainly institution-based studies. Therefore, this study assess the prevalence of MetS among adults who are residing in Gondar city using Adult Treatment Panel III (ATP III) and the International Diabetes Federation (IDF) assessment tool. The findings are imperative to developing and strengthening national NCD prevention and control programs.

### Methods

This study was conducted in Gondar city Northwest Ethiopia in 2018. It employs a community-based cross-sectional study design among 3,227 individuals 18 years of age or older. Data was collected using the WHO stepwise tool, lipid profile, blood pressure (BP), waist circumference (WC) body mass index (BMI), fasting blood glucose levels (FG), and anthropometric measurements. The prevalence estimation was made along with a 95% confidence interval (CI). The Kappa statistic was used to analyze the statistical agreement between ATP III and IDF definitions of the MetS. Stratified analysis was also performed for description and analysis components using ATP III and IDF as an outcome.

### Result

Of the total study participants (3227), 3059 (94.8%) were included in the final analysis and 52.5% were female. The mean (±SD) age of the study participant was 40.8 years (16.2 ±SD). The overall prevalence of MetS using ATP III was 11.2% [95%CI: 10.1, 12.3] and using IDF was 11.9% [95%CI: 10.8, 13.2]. The sex-specific proportion was high in females rather than males irrespective of the criteria. The overall level of agreement between ATP III and IDF prevalence was 91.7% and the Kappa statistics was 0.594. Older age, low-density

**Funding:** This study was partially funded by University of Gondar for data collection. No additional external funding was received for this study.

**Competing interests:** The authors have declared that no competing interests exist.

lipoprotein cholesterol, body mass index, being female, born in an urban area, consumption of an alcoholic drink in the preceding 30 days, and non-fasting practice was significantly associated with MetS.

## Conclusion and recommendation

There was a higher prevalence of metabolic syndrome among females than males irrespective of metabolic syndrome diagnostic criteria. This also shows good agreement between ATP III and IDF. Being female, urban birthplace, frequent alcohol consumption in the last 30 days, and non-fasting practice are factors associated with higher rates of metabolic syndrome. Hence, awareness campaigns, physical exercise, and nutrition education intervention should be undertaken to promote health behavioral practice.

## Introduction

Metabolic syndrome (MetS) refers to a condition that inhabits three of the five following medical conditions which include central obesity, decreased high-density lipoprotein cholesterol, elevated triglycerides, elevated blood pressure, and hyperglycemia [1, 2]. With its prevalence in becoming a major public health problem worldwide, it is clear that the burden of metabolic syndrome is rapidly increasing the rates of non -communicable diseases (NCD) [2]. The low- and middle-income countries (LMC) of the world are the most affected by these diseases which will have substantial social, economic, and health consequences [3–5]. Internationally the prevalence is different among regions, one of the lowest prevalence was observed in China among children (0.7%) (0.7%) [6]. On the other hand high prevalence was observed in Indus Hospital, Karachi, Pakistan (97.5%) [7] and in Palestinian refugees (among obese and overweight participants) (69.4%) [1].

Fast-paced urbanization and modernization, changes related to unhealthy lifestyles which are coupled with climatic changes give rise to intermediate-risk factors to cardiovascular diseases (CVD) such as Metabolic syndrome, unfavorable lipid profiles, and obesity [8–10]. Findings have shown that basal metabolic rate is consistently affected by climatic change, metabolic defects (that are adaptations to lifestyle), diet and climates experienced by human populations over long-term evolutionary time. The biological processes that influence tolerance to climatic extremes are likely to play important roles in the pathogenesis of common metabolic disorders, such as obesity, and dyslipidemia. The combination of diets high in fructose and salty foods, increasing temperatures, and decreasing water availability in certain places can accelerate metabolic syndrome.

With the current rapid urbanization, people in developing countries are increasingly eating processed foods and tobacco, alcohol, and junk food which are known to increase the risk for metabolic syndrome. Currently, more and more labor-saving devices that keep people increasingly indoors are being created which could be an explanation for the increases in obesity and CVD [11, 12].

Ethiopia is facing a dual burden, on the one hand with a high number of infectious diseases and under-nutrition while on the other hand having a significantly increasing rate of NCDs and obesity. In addition to the increasing urbanization and associated adoption of an urban life style, the concept of climate change has also impacted the dietary practice and lifestyle of Sub-Saharan Africa by altering the nutritional status, infectious disease occurrence and

development of NCDs. This led to an update of the government's strategic plan which allowed it to, at least for the better, combat NCDs in the country [13]. This burden on multiple fronts, which is far beyond what is experienced in developed countries, imposes immense pressure on the already struggling preventive and curative health care system of the country. Most of the previous studies were focused on institutional-based facilities (hospital) and the study participants were adult patients who visited the health facility for another comorbidity which can ultimately directly or indirectly affect the metabolic level of the study participants [14, 15].

Since the level of metabolic syndrome and its components are expected to be higher among study participants recruited from health facilities compared to the normal population found in the community (due to their pre-existing conditions), those studies could not give the true picture of the magnitude of metabolic syndrome and its distribution in the general population or community [3]. Also, very little evidence exists on the epidemiology of metabolic syndrome at a community level in Ethiopia, and no adequate attention has been given to this epidemic and the care management of metabolic syndrome as it competes for system resources allocated to infectious diseases.

Previous studies done in Ethiopia have shown a large disparity in the magnitude of metabolic syndrome. One of the studies done in 2015 among 9 regions and two city administrations including Addis Ababa, showed a prevalence of metabolic syndrome of 4.8%. Another population-based survey conducted in Jimma from 2008–2009 showed 10.7% and a recent study in Addis Ababa showed a prevalence of 20.3% [3, 16, 17]. Previous studies used different criteria and numerous cases used either ATP III or IDF to ascertain the metabolic syndrome as the diagnostic criteria. None of the studies reported the distribution using age stratification with a sufficient sample size. Since the use of various diagnostic criteria for different populations and studies could confound the comparison result, it would be ideal to use both Adult Treatment Panel III (ATP III) and International Diabetes Federation (IDF) diagnostic criteria in the same population to show the extent of agreements or differences between the diagnostic criteria. Hence, this study assessed metabolic syndrome using both criteria (ATP III and IDF) and found criteria agreements for the same study population at the community level which will show a better picture of the distribution of metabolic syndrome with different factors [1]. Moreover, previous studies seem to have gaps in terms of lifestyle and anthropological perspectives which are needed to explain the reasons, deterrence, and control of metabolic syndrome, especially in Ethiopia, where health outcomes are highly dependent on dietary practice [18, 19]. Therefore, establishing community-based baseline data on the prevalence of metabolic syndrome using Adult Treatment Panel III (ATP III and IDF) is imperative to developing and strengthening national NCD prevention and control programs.

## Materials and methods

### Study context

This study was conducted in Gondar town Northwest Ethiopia, located in East Africa. Ethiopia is Africa's second-most populous country with an estimated population of 109,224,414 in 2018 according to the 2019 revision of the World Population Prospects, of which 18% live in urban areas. Gondar city is a densely populated historical city of the region with an estimated urban population of 356,290.

### Study design

This study implemented a community-based cross-sectional study design.

## Study population

All individuals in the study were 18 years of age or older and resided in Gondar city for at least six months. All permanent residents in the study area, aged 18 years or older, were eligible to participate in the study.

## Sample size and sampling

The sample size for the study was determined by assuming difference in the prevalence of metabolic syndrome across different age, and sex categories. Most of the studies in our region are limited to a pooled estimate using single population proportion sample size formula, for this reason, this study used a 50% prevalence with a 95% confidence interval and a 5% margin of error for each age group (6 groups) [20]. This study also considered a 5% non-response rate. After acquiring the minimum sample size stratification for age and sex (four age categories multiplied by two (for male and female), the calculated sample size was 3,227 individuals. A detail of the method for this study was published elsewhere [21].

This study used a two-stage simple random sampling strategy to select the household. Initially, four Sub-Cities (administrative districts) were selected using simple random sampling from the overall 12 urban Sub-Cities (after obtaining the list from the Gondar city administration). Then, household were selected within each sub-city using the systematic random sampling technique. Finally, in the cases where there was more than one eligible person in the household the study used a lottery method to select the study participant.

## Data collection

Data was collected by trained field workers that included local community enumerators, laboratory technicians, and nurses. They collected the data by going house-to-house. To ensure the quality of the interview, data collectors were trained by the principal investigator and later on random checks were carried out by field supervisors and the principal investigator. Data was collected by interviewing eligible subjects using a structured questionnaire. House-to-house data collection was performed by trained Laboratory technicians and Health extension nurses.

## Measurements

High blood pressure was classified as hypertension with systolic blood pressure (SBP) of ≥140mmHg and /or diastolic blood pressure of (DBP) ≥ 90 mmHg. Anthropometric measurements were taken using standardized techniques and calibrated equipment. Blood pressure (BP) was measured using a digital measuring device after resting for at least five minutes prior. Finally, biochemical tests (fasting blood glucose levels, triglyceride, LDL, HDL, and total cholesterol test) were carried out. Impaired fasting blood glucose levels (IFG) were identified if the fasting blood glucose levels were between 110 and 125 mg/dl. Blood samples were collected from each participant by a trained laboratory technician following aseptic techniques. The blood samples were taken to the hospital laboratory for chemistry analyses. Anthropometric measurements were taken using WHO stepwise standardized techniques and calibrated equipment. Subjects were weighed to the nearest 0.1 kg in light indoor clothing and bare feet or with stockings. Participant's height was measured using a stadiometer. This is done when the subject stands in erect posture without shoes and recorded to the nearest 0.5cm. Measures were taken two times, and the average was considered in the analysis. Waist girth was measured by placing a plastic tape to the nearest 0.5 cm, horizontally midway between the 12th rib and the iliac crest on the mid-axillary line. The hip was measured horizontally on the greater trochanters. Waist circumference (WC) was categorized as low risk if it was less than 94 cm for men,

and less than 80 cm for women; and high risk if it was 94 cm or more for men, and 80 cm or more for women. Body mass index (BMI) was calculated as the ratio of body weight in kilograms to the square of body height in meters. BMI was used to define underweight (BMI < 18.5), normal (18.5 ≤ BMI < 25.0), overweight (25.0 ≤ BMI < 30.0), and obese (BMI ≥ 30) adults. Current alcohol consumption was assessed by asking participants to respond by ticking "Yes /No" to the question, "have you consumed any alcoholic drink, such as beer, wine, Tela, Tej, local Areki, fermented cider in the last 30 days?". Data about current smoking was found out by asking participants to respond in the same manner to the question, "Do you currently smoke any tobacco products, such as cigarettes?" Moderate physical activity was considered as "Yes" for those participants who walked at least for 10 minutes continuously daily [21].

## Dependent variable

To measure the dependent variable we use IDF Definition: By the IDF criteria, subjects were classified as having metabolic syndrome if participants had abdominal obesity (defined as waist circumference of ≥94 cm for men and ≥80 cm women) plus two of any of the following risk factors: (1) raised TG level (≥150 mg/dL); (2) reduced HDLC (<40 mg/dL in males and <50mg/dL in females); (3) raised blood pressure (systolic BP ≥130 or diastolic BP ≥85mmHg) or treatment of previously diagnosed hypertension; and (4) raised FG (≥100mg/dL). Under the ATP III criteria, subjects were classified as having metabolic syndrome if participants had three or more of the following risk factors: (1) abdominal obesity (waist circumference >102 cm in males and >88 cm in females); (2) hypertriglyceridemia (TG ≥150mg/dL); (3) reduced HDL-c (<40 mg/dL in males and <50mg/dL in females); (4) high BP (≥130/85mmHg); and (5) FG (≥100mg/dL).

## Data management and analysis

Double data entry procedures were done using the EPI Info 7 statistical software. Distributive statistics were performed using tables and graphs. The prevalence estimation was made along with a 95% confidence interval (CI). The Kappa (kappa measure of interrater agreement Scale reliability coefficient) statistic was used to analyze the statistical agreement between ATP III and IDF definitions of the metabolic syndrome. Stratified analysis was also performed to describe metabolic syndrome using ATP III and IDF for multivariate analysis. Independent variables were selected based on a cut of point p-value <0.20 was included for the multivariable model. Binary logistic regression was applied to identify factors associated with metabolic syndrome. Variables having a p-value of < 0.20 in the univariate analysis (Chi-square tests) was used as a criteria in the multivariable logistic regression model to control confounding effects, and the results were considered statistically significant at P-value ≤ 0.05. Details of the procedures were published elsewhere [21].

## Ethical statement

Ethical clearance was obtained from the University of Gondar ethical review board. Subjects who volunteered to participate in the study were included after signing a written informed consent. They were informed of their rights to withdraw from the study at any stage. For the sake of privacy and confidentiality, no personal identifiers such as names were collected.

## Result

Of the total study participants, 3,059 (94.8%) were included in the final analysis and 52.5% were female. The mean (±SD) age of the study participants was 40.8 years (16.2 ±SD), the majority (53.12) of them did not attend formal education and 1,728 (56.5%) of the study participants were married. Only 520 (17%) were government and private employees. Socio-demographic characteristics of the study participants are presented in Table 1.

**Table 1. Socio- demographic and clinical characteristics of the study population by sex in a community-based survey among Gondar city residents who were ≥18 years old, Northwest Ethiopia.**

| Variable | Male n(%) | Female n(%) | Total n(%) |
|---|---|---|---|
| **Age in Years** | | | |
| 18–24 | 264 (47.2) | 295 (52,8) | 559 (18.3) |
| 25 to 34 | 305 (46.8) | 347 (53.2) | 652 (21.3) |
| 35 to 44 | 253 (43.4) | 330 (56.6) | 583 (19.1) |
| 45 to 54 | 236 (46.3) | 274 (53.7) | 510 (16.7) |
| 55 to 64 | 163 (42.3) | 223 (57.8) | 386 (12.6) |
| 65 and above | 185 (50.1) | 184 (49.9) | 369 (12.1) |
| **Location Birth** | | | |
| Urban | 633 (48.1) | 684 (51.9) | 1317 (43.1) |
| Rural | 773 (44.4) | 969 (55.6) | 1742 (56.9) |
| **Education status** | | | |
| Unable to read and write | 193 (41.2) | 276 (58.8) | 469 (15.3) |
| Can read and write | 590 (47.6) | 649 (52.4) | 1239 (40.5) |
| Primary school | 146 (53.7) | 126 (46.3) | 272 (8.9) |
| Secondary school | 117 (70.5) | 49 (29.5) | 166 (5.43) |
| Diploma | 13 (61.9) | 8 (38.1) | 21 (0.69) |
| Degree and above | 347 (38.9) | 545 (61.1) | 892 (29.16) |
| **Marital Status** | | | |
| Single | 483 (55.8) | 383 (44.2) | 866 (28.3) |
| Married | 832 (48.2) | 896 (51.8) | 1728 (56.5) |
| Separated | 21 (17.1) | 102 (82.9) | 123 (4.0) |
| Divorced | 29 (23.8) | 93 (76.2) | 122 (3.9) |
| Widowed | 41 (18.6) | 179 (81.4) | 220 (7.2) |
| **Religion** | | | |
| Orthodox | 1230 (45.5) | 1473 (54.5) | 2,703 (90.9) |
| Muslim | 129 (48.9) | 135 (51.1) | 264 (8.9) |
| Protestant | 4 (66.7) | 2 (33.3) | 6 (0.2) |
| **Main work type over the last 12 months** | | | |
| Government employee | 201 (56.8) | 153 (43.2) | 354 (11.6) |
| Private employee | 100 (60.2) | 66 (39.8) | 166 (5.4) |
| Personal Job | 715 (58.4) | 509 (41.6) | 1224 (40.0) |
| Non paid Job | 5 (11.1) | 40 (88.9) | 45 (1.47) |
| Student | 162 (52.3) | 148 (47.7) | 310 (10.1) |
| Home worker | 3 (5.3) | 54 (74.7) | 57 (1.9) |
| Retired | 53 (70.7) | 22 (29.3) | 75 (2.4) |
| Able to work | 132 (20.0) | 528 (80.0) | 660 (21.6) |
| Unable to work | 35 (20.8) | 133 (79.2) | 168 (5.5) |

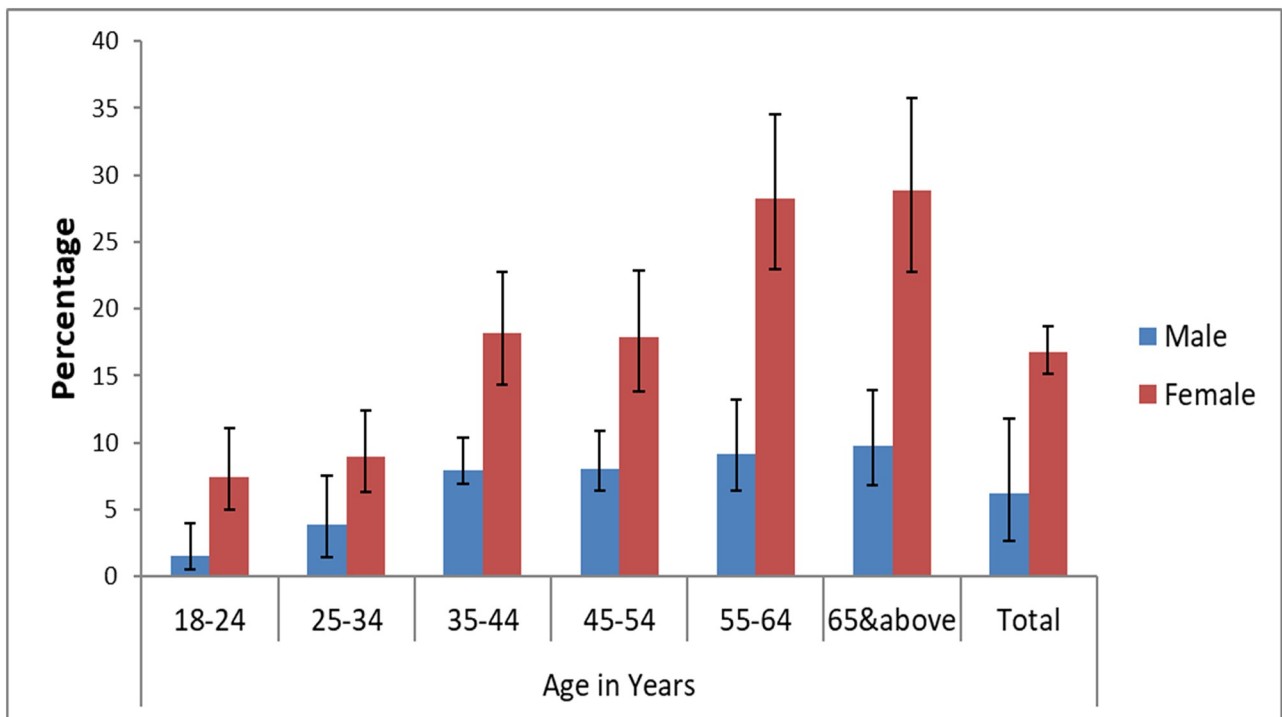

**Fig 1. Distribution of IDF criteria based metabolic syndrome by age and sex among ≥18 years old Gondar city residents, Northwest Ethiopia.**

The overall prevalence of metabolic syndrome using ATP III was 11.2% [95%CI: 10.1, 12.3] and that of IDF was 11.9% [95%CI: 10.8, 13.2]. The sex-specific proportion was high in females using ATP III and using IDF was 16.8% in females and 6.3% in the male study population.

The prevalence of metabolic syndrome increased with the increasing of age (1.9% in the age group of 18 to 24 years to 20.2% in the age group of 45 to 54 years). Details of the metabolic syndrome profile is presented in Figs 1 and 2.

The proportion of metabolic syndrome was slightly higher among diabetic subjects (37.2%) using both IDF and ATP III criteria compared to its proportion among subjects with high blood pressure (27.4% using ATP III and 26.5% using IDF criteria). Biochemical and anthropometric characteristics of the study population is presented in Table 2.

The distribution of high triglyceride, high total cholesterol, and high LDL cholesterol by age and gender are presented in Figs 2–4.

Details of percentage distribution of metabolic syndrome by ATP III and IDF criteria were presented in Fig 5.

The overall level of agreement between ATP III and IDF prevalence was 91.7%, which does not take into account chance agreement. On the other hand, the Kappa statistics was 0.594 with the agreement deference that could be due to chance. In terms of sex, Kappa is better among females (0.589) than males (0.507). The risk factor for cardiovascular disease, in general, is presented in S1 Table.

Variables included in the univariate analysis were age, sex, residence, religion, education status, marital status, main work type, current smoker, current alcohol intake, vegetable and fruit intake, fasting habit, doing moderate exercise and history of TB. However only those whose p-value that was less than 0.20 were included in the final model. Using the IDF criteria the multivariable logistic regression model showed that for every unit increase in age by one

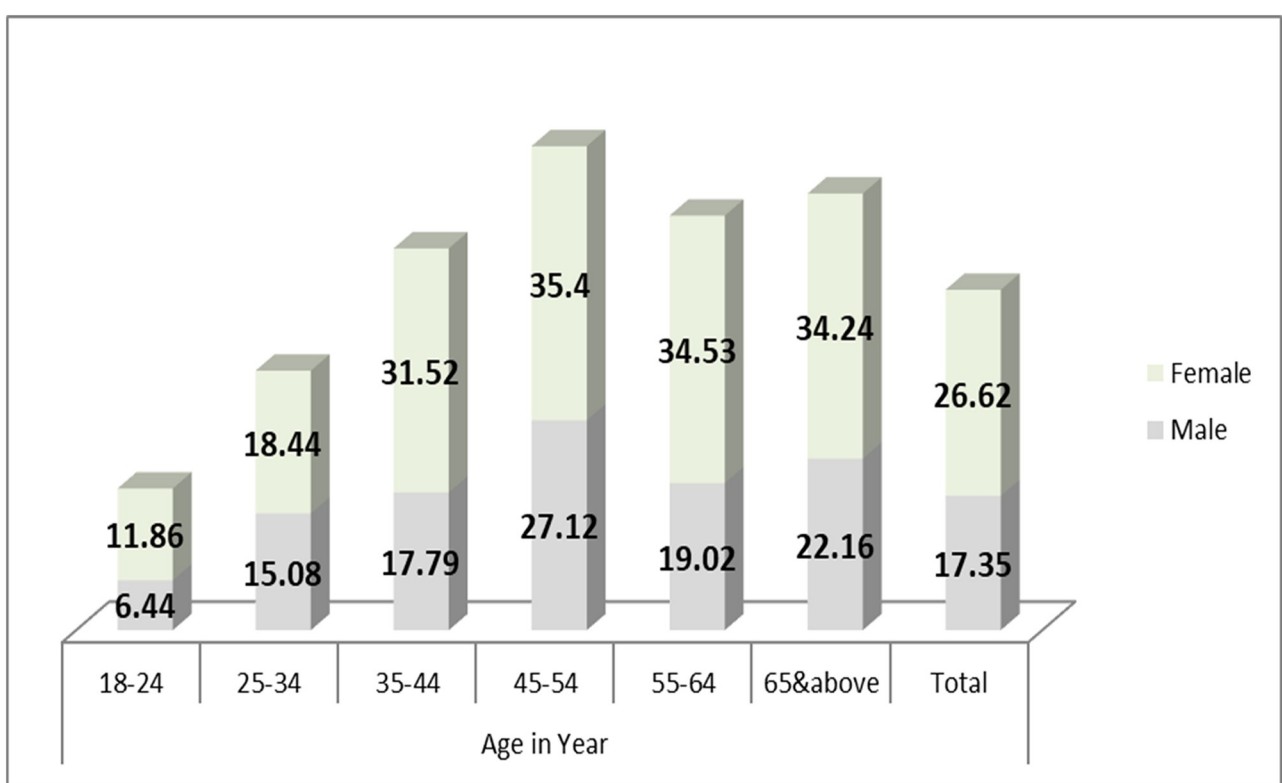

**Fig 2. Distribution of high total cholesterol level by age and sex among ≥18years old Gondar city residents, Northwest Ethiopia.**

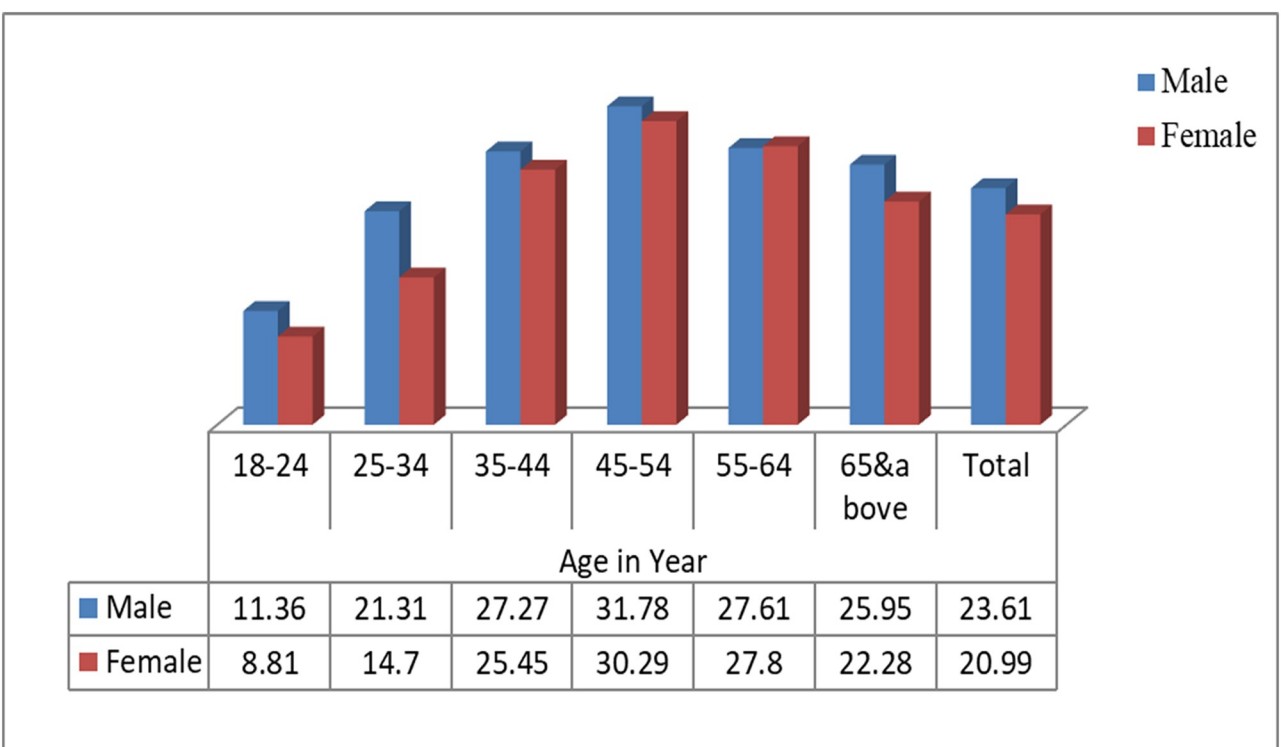

| | 18-24 | 25-34 | 35-44 | 45-54 | 55-64 | 65&above | Total |
|---|---|---|---|---|---|---|---|
| ■ Male | 11.36 | 21.31 | 27.27 | 31.78 | 27.61 | 25.95 | 23.61 |
| ■ Female | 8.81 | 14.7 | 25.45 | 30.29 | 27.8 | 22.28 | 20.99 |

**Fig 3. Distribution of high triglyceride level by age and sex among ≥18 years old Gondar city residents, Northwest Ethiopia.**

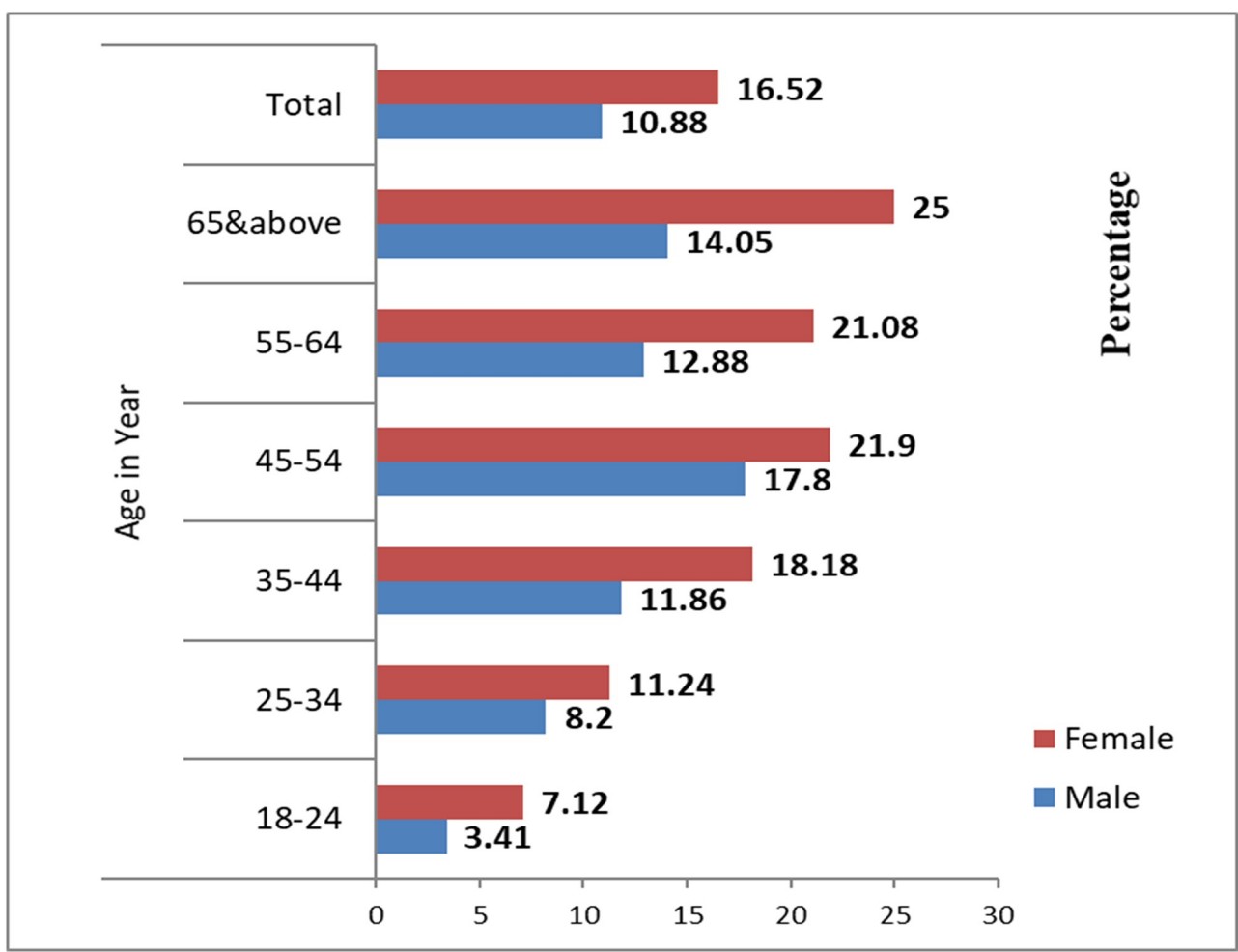

**Fig 4. Distribution of high LDL cholesterol level by age and sex among ≥18 years old Gondar city residents, Northwest Ethiopia.**

year there was an increase in the prevalence of metabolic syndrome by 1% [AOR = 1.01%; 95% CI:1.001, 1.017]. A unit increase in low-density lipoprotein cholesterol in milligram /deciliter will increase the prevalence of metabolic syndrome by 0.6%, [AOR = 1.006%; 95%CI:1.002, 1.009]. Similarly, a unit increase in body mass index (kg/m$^2$) of a person will increase the

**Table 2. The mean (±SD) Biochemical and anthropometric characteristics of the study population in a community-based survey among Gondar city residents who were ≥18 years old, Northwest Ethiopia.**

| Variable | Mean | Standard deviation (±SD) |
|---|---|---|
| Waist Circumference (cm) | 71.48 | 23.15 |
| Hip Circumference (cm) | 91.33 | 19.08 |
| Systolic Blood pressure (mmHg) | 125.28 | 19.76 |
| Diastolic Blood pressure(mmHg) | 78.51 | 11.37 |
| Triglyceride (mg/dL) | 116.58 | 87.44 |
| Low density lip-protein cholesterol (mg/dL) | 89.07 | 39.66 |
| High density lip-protein cholesterol (mg/dL) | 48.50 | 19.84 |
| Total cholesterol (mg/dL) | 165.80 | 48.37 |
| Fasting blood glucose (mg/dL) | 80.71 | 28.25 |

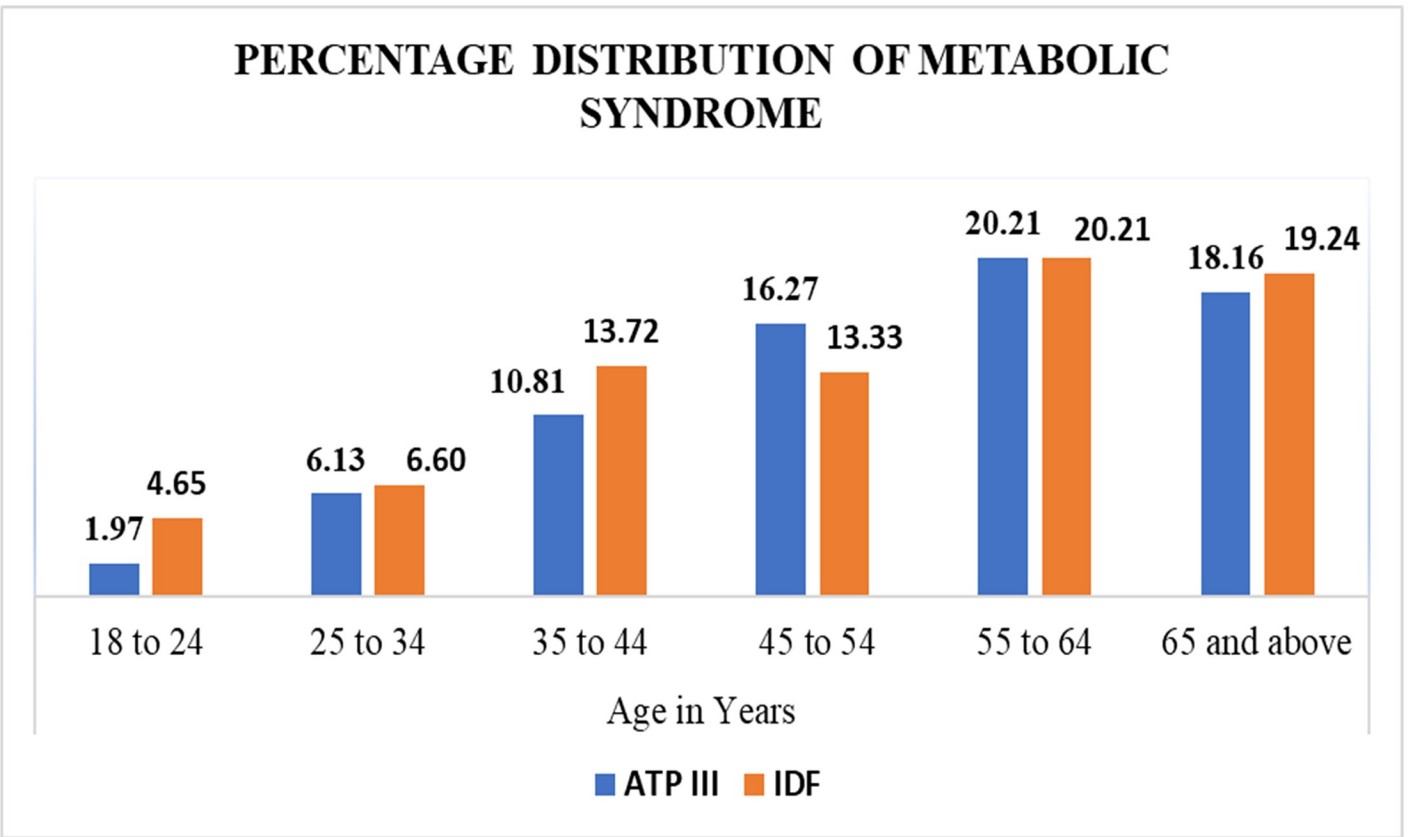

**Fig 5. Distribution of metabolic syndrome by ATP III and IDF criteria among different age group in Gondar city residents, Northwest Ethiopia.**

prevalence of metabolic syndrome by 18% [AOR = 1.18%; 95%CI:1.1.14, 1.23] using IDF classification. The occurrence of metabolic syndrome was more than 2 times higher among females than males [AOR = 2.06; 95%CI:1.47, 2.90]. The occurrence of metabolic syndrome is 50% higher among people who were born in an urban area than their rural counterparts [AOR = 1.52; 95%CI: 1.11, 2.08]. The odds of metabolic syndrome were more than 3 times higher among subjects who had separated/widowed than that of single individuals [AOR = 3.15; 95%CI: 1.79, 5.54]. The occurrence of metabolic syndrome was more than 48% higher among non-fasting people than frequently fasting religious people at the time of fast (regardless of the classification) AOR = 1.48; 95%CI: 1.048, 2.09] (see Table 3).

## Discussion

This study is one of the very few studies to assess metabolic syndrome in the community, and we have uncovered a higher prevalence of metabolic syndrome among females rather than males irrespective of metabolic syndrome diagnostic criteria. The proportion of metabolic syndrome was slightly higher among subjects with high fasting blood glucose than those with high blood pressure cases. The overall level of agreement between ATP III and IDF prevalence was high (91.7%). In terms of sex, Kappa is better among females (0.589) than males (0.507) (91.7%) [22]. The finding is in line with a previous study done in Ethiopia (K 0.54) [23] and in Ghana [24]. Low-density lipoprotein cholesterol, body mass index, being female, urban birthplace, frequent alcoholic consumption for the last 30 days, and frequently fasting are factors associated with high metabolic syndrome irrespective of the criteria used.

**Table 3. Multivariable analysis of factors associated with metabolic syndrome by ATPIII and IDF classification in Gondar city, Northwest Ethiopia (2016).**

| Variables | ATP III MetS | IDF MetS |
|---|---|---|
| | AOR [95%CI] | AOR [95%CI] |
| **LDL cholesterol** | 1.004 [1.00, 1.008] * | 1.006 [1.002, 1.009] ** |
| **BMI** | 1.21 [1.16, 1.25] ** | 1.18 [1.14, 1.23] ** |
| **Age in year** | 1.016 [1.007, 1.025] ** | 1.01 [1.003, 1.017] ** |
| **Sex** | | |
| Male | 1.00 | |
| Female | 2.74 [1.88, 3.98] ** | 2.06 [1.47, 2.90] ** |
| **Location of birth place** | | |
| Rural | 1.00 | |
| Urban | 1.45 [1.04, 2.03] * | 1.52 [1.11, 2.08] * |
| **Marital status** | | |
| Single | 1.00 | |
| Married | 2.04 [1.16, 3.59] * | 1.62 [0.99, 2.64] |
| Separated/Widowed | 3.16 [1.66, 5.99] ** | 3.15 [1.79, 5.54] ** |
| **Have you ever consumed any alcoholic drink** | | |
| No | 1.00 | |
| Yes | 1.23 [0.68, 2.22] | 1.27 [0.73, 2.23] |
| **Do you fast** | | |
| Yes | 1.00 | |
| No | 1.22 [0.84, 1.78] | 1.48 [1.04, 2.09] * |

*P-value < 0.05;

** P-value < 0.001.

The prevalence of metabolic syndrome in this study is consistent with a study done in Addis Ababa among adult bank workers (10%) [23] and slightly higher than the findings in community-based studies done in an Indian rural community and residents of Mizan-Aman Ethiopia [16, 25, 26]. Depending on the study site, the prevalence of metabolic syndrome is expected to be high in the urban area which has been demonstrated in other studies [9, 27]. The prevalence of metabolic syndrome in this study is lower compared to findings of studies done in Ghana [24] and rural Uganda [24]. Similarly, the prevalence of metabolic syndrome in the study was lower than reports in Jimma, South West Ethiopia, and that of the University of Gondar Hospital [19, 28]; but those two studies were conducted among individuals with mental health problems. This finding is lower than the report from other studies conducted in Mekelle, Northern Ethiopia (20.8%), and Haramaya University employees (20.1%) in Eastern Ethiopia; and, it is also lower than the pooled prevalence (34.9%) done in Ethiopia [14, 15].

The observed difference in the prevalence of those studies could be attributed to the fact that most of them were facility-based and the participants were more likely to have a higher chance of metabolic syndrome development due to the comorbid psychiatric illness compared to those in the general population at the community [2, 3]. Additionally, the current study has included adults (18years and older), unlike those previous studies which mainly included older adults (mostly 35 and above older). The age spectrum difference between our study and the previous study may contribute to the discrepancies in the overall prevalence.

Previous studies revealed that the incidence of cardiovascular disease (CVD) is high with older age; likewise, the prevalence of metabolic syndrome showed an increasing trend with

increasing age in this study. The finding is also consistent with reports from China and Brazil [25, 29]. The prevalence of metabolic syndrome was found to increase among participants of advanced age, obesity, low physical activities, and unhealthy dietary practice. Such findings are warranting an early intervention to halt the progression of cardiovascular disease (CVD) as metabolic syndrome is a precursor of CVD in older age individuals.

During a woman's lifetime, there are normal events such as puberty, pregnancy, and menopause, which are related to alterations in energy homeostasis and increased body fat due to hormonal effects. Furthermore, conditions of severe metabolic stress and energy imbalance are commonly linked to alterations in metabolic syndrome. In line with this, our observation showed a high prevalence of metabolic syndrome among females. This finding is similar to studies done on African women [30, 31], Iranian women [32], and black African American women in the USA which show a higher prevalence of metabolic syndrome among women than men [33, 34]. This finding could imply that metabolic syndrome is a more prominent risk for developing cardiovascular disease (CVD) among women than men and has been suggested by previous studies [35, 36]. In this regard, previous studies have also indicated the high relevance and burden of metabolic syndrome among females and the need for preventive intervention to delay or prevent its occurrence [37, 38].

In our findings, the proportion of metabolic syndrome was slightly higher among participants who also had high fasting blood glucose. Studies have reported that metabolic syndrome is a cluster of glucose intolerance, hypertension, dyslipidemia, and central obesity and it predicts diabetes independently of other factors [24, 33, 39, 40]. People with metabolic syndrome are more likely to develop diabetes and this will increase the risk of developing cardiovascular disease.

A high proportion of the study participants were found to have low HDL protein which is consistent with a finding by another study done in Africa [32]. The most frequent abnormalities like low HDL-cholesterol, high LDL-cholesterol, and high total cholesterol are precursors of metabolic syndrome and cardiovascular diseases.

Urban birth-place was significantly associated with metabolic syndrome, which is in line with a systematic review and meta-analysis done in sub-Saharan Africa [29]. This finding is consistent with other reports and findings that share that sedentary lifestyle (unhealthy dietary practice and lack of exercise), predominantly high in the urban area [40], is a plausible explanation to the observed high metabolic syndrome in an urban adult population. The study revealed alcohol consumption 30 days prior to testing has a significant association with metabolic syndrome, and this finding is in line with other studies [31, 41, 42].

In addition, the frequency of fasting was associated with metabolic syndrome. In this finding individuals who were fasting frequently had low metabolic syndrome compared with non-fasting participants. Most Ethiopian Orthodox Christianity followers, which are (90.9%) of our study participants, usually ate vegetables, cereals, and other vegan foods during fasting seasons (after not eating for 15 to 18 hours) and two days a week in the non-fasting seasons. Previous studies showed that intermittent fasting could protect against metabolic syndrome [43–45]. In most cases, prolonged fasting and avoidance of dairy products at the time of religious fasting is common in the study area. The short-term improvements in some of the components of metabolic syndrome could be achieved by narrowing participants' eating periods by time-restricted eating. In addition, vegan foods will reduce the chance of developing metabolic syndrome [43].

Concerning the limitation of this study we found that the cross-sectional study design, was a hindering factor in making the temporal association between the factors found to be associated with metabolic syndrome in this study. Children and young adolescents were not included in the study. Biomedical and anthropometric measurements are prone to errors;

however, this study was conducted with effective training and frequent supervision and control mechanisms. Moreover, a checklist was developed before the actual training. The checklist included equipment calibration, and standardization of procedures that would be dealt during data collection and analysis which would minimize the possibility of errors. This study used a large sample size, including both young and old participants, for that reason, this study is adequately powered to show the magnitude of metabolic syndrome across the spectrum of age. In addition, most of the previous studies of metabolic syndrome were either hospital-based or targeting a specific patient population. This study has revealed the bigger picture of the problem by conducting the study on the general community.

## Conclusion and recommendation

The prevalence of the metabolic syndrome is high in the urban community and relatively higher among females and increases with age. Though less prevalent than the general population, a significant proportion of young adults have metabolic syndrome. And a large proportion of participants have a low level of protective cholesterol, HDL. Low-density lipoprotein cholesterol, body mass index, being female, having an urban birthplace, frequent alcohol consumption for the past 30 days, and non-fasting are factors associated with metabolic syndrome. We recommend both general and then targeted health education and health promotion interventions to be coordinated and employed by the relevant governmental and non-governmental stakeholders to curb the emerging burden of non-communicable diseases in the resource-limited setting before it threatens the entire healthcare system.

## Supporting information

**S1 Table. Characteristics of CVD risk factors by sex and age among Gondar city residents who were ≥18 years old, Northwest Ethiopia.**
(DOCX)

**S1 File. Study tool.**
(DOC)

## Acknowledgments

The authors are very grateful to the participants for committing their time and providing information and specimens. The authors would like to thank the University of Gondar for the full support and facilitation to allow us to conduct the study.

## Author Contributions

**Conceptualization:** Solomon Mekonnen Abebe, Abayneh Girma Demisse, Shitaye Alemu, Bewketu Abebe.

**Data curation:** Solomon Mekonnen Abebe, Abayneh Girma Demisse.

**Formal analysis:** Solomon Mekonnen Abebe, Abayneh Girma Demisse, Bewketu Abebe.

**Investigation:** Solomon Mekonnen Abebe, Abayneh Girma Demisse, Shitaye Alemu, Bewketu Abebe, Nebiyu Mesfin.

**Methodology:** Solomon Mekonnen Abebe, Shitaye Alemu, Bewketu Abebe, Nebiyu Mesfin.

**Project administration:** Abayneh Girma Demisse.

**Software:** Solomon Mekonnen Abebe, Abayneh Girma Demisse, Nebiyu Mesfin.

**Supervision:** Solomon Mekonnen Abebe, Abayneh Girma Demisse, Nebiyu Mesfin.

**Validation:** Solomon Mekonnen Abebe, Abayneh Girma Demisse.

**Visualization:** Abayneh Girma Demisse, Nebiyu Mesfin.

**Writing – original draft:** Solomon Mekonnen Abebe, Abayneh Girma Demisse, Bewketu Abebe, Nebiyu Mesfin.

**Writing – review & editing:** Solomon Mekonnen Abebe, Shitaye Alemu, Bewketu Abebe, Nebiyu Mesfin.

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
