## [Decision Letter · Decision Letter 0]

11 May 2021

PONE-D-21-02096

Magnitude of Metabolic Syndrome in Gondar Town, northwest Ethiopia: A community-based cross-sectional study

PLOS ONE

Dear Dr. Solomon Mekonnen Abebe,

Thank you for submitting your manuscript to PLOS ONE. After careful consideration, we feel that it has merit but does not fully meet PLOS ONE’s publication criteria as it currently stands. Therefore, we invite you to submit a revised version of the manuscript that addresses the points raised during the review process.

Authors should address issues pointed out by reviewers, mainly:

-English should be extensively revised.

-Introduction should be improved.

-Methodology needs important clarifications.

-Results and Discussion section should be more focused on the main results of the study.

We look forward to receiving your revised manuscript.

Kind regards,

Pedro Tauler, Ph.D.

Academic Editor

PLOS ONE

Journal Requirements:

[The authors are very grateful for the funding provided by University of Gondarto conduct this study with full support,includingtraining every study participant at each step of the study.]

 [No The funders had no role in study design, data collection and analysis, decision to publish, or preparation of the manuscript]

Reviewers' comments:

Reviewer's Responses to Questions

**Comments to the Author**

1. Is the manuscript technically sound, and do the data support the conclusions?

Reviewer #1: Partly

Reviewer #2: Partly

Reviewer #3: Partly

2. Has the statistical analysis been performed appropriately and rigorously? 

Reviewer #1: N/A

Reviewer #2: No

Reviewer #3: No

3. Have the authors made all data underlying the findings in their manuscript fully available?

Reviewer #1: Yes

Reviewer #2: No

Reviewer #3: Yes

4. Is the manuscript presented in an intelligible fashion and written in standard English?

Reviewer #1: No

Reviewer #2: No

Reviewer #3: No

5. Review Comments to the Author

Reviewer #1: Thanks to the authors

It is better to correct and improve the English writing of the text.In the methodology section, the method of measuring the sample size is not completely clear

Please see the word's attached file.

Reviewer #2: Thanks for inviting me to review this manuscript entitled "Magnitude of Metabolic Syndrome in Gondar Town, northwest Ethiopia: A community-based cross-sectional study". This manuscript reported a community-based cross-sectional study in Ethiopia (n=3059), which provided a picture for the prevalence of MetS in East Africa. Please see the attachment froe detailed comments.

Reviewer #3: This study assesses the Magnitude of Metabolic Syndrome in Gondar Town, northwest Ethiopia which is a community based cross-sectional study. Though the concept as whole is interesting, however, there are some shortcomings in the present study which are worth to look into and address:

1. The introduction section does not amply describe a need to conduct a prevalence study on MetS on the said population. The authors themselves hinted on various such studies done already in this population and as per them, those seemed to have gaps in terms of lifestyle and anthropological perspectives; however, the authors in this study lacked to mention details on how those gaps were addressed in this study.

2. The other concern regarding this study is the methodology used as the sampling design does not adequately cover the population due to the random lottery method used. The survey can also fail to cover the population as it does not categorize the people living in both standard and non-standard dwellings. This is an important factor for any community- based study.

3. Other issue about the methodology is the ATPIII criteria which is best predictor of the cardiovascular events but in elderly people. The conclusion drawn from the study thus cannot be extrapolated to other age-groups. Besides, the sample size calculation shown in the methodology section to select a representative sample for prevalence studies (especially when prevalence’s of >30% MetS have been reported in some studies) for a population of about 109 million needs revisiting.

4. The measurement section is a little confusing to the readers as the first sentence first sentence itself states that WHO and IDA criterion were used to classify hypertension while as per the abstract and rest of the manuscript, NCEP ATPIII or IDF were used to classify all the components of MetS. Also, the way of presenting the questionnaire details in the last paragraph of measurement section needs revisiting and should be presented in a more scientific manner.

5. The result section should have been presented as and according to the data given in the tables and highlighting on the main findings which needed to be discussed in the discussion section. Also, the results should have been presented in accordance with the study objective which was to derive at the prevalence of MetS in the said population. MetS, according to the criteria used, has components like “central obesity”, “hypertension”, “hypertriglyceridemia”, “low HDL-cholesterol” and “hyperglycemia” and thus focus should have been on these components and the full MetS rather than describing hypercholesteromia, high LDL-C, etc.

6. The discussion lacks the details of the defense of the observations and findings in the current study. For example, the initial statements in the discussion relates the findings in the current study in line with a study (reference # 20) while the sample size in that study was 314 and the prevalence of MetS observed was 59.9% and 70.1% according to IDF and NCEP criteria’s respectively. Besides, the authors struggled in explaining the reason for such a low prevalence of MetS (as we have numerous reports of much higher prevalence of MetS in the general population worldwide).

7. Lastly, there are numerous grammatical mistakes in the entire manuscript which should be looked at by the authors seriously before resubmission. Besides, the statistics done should be revisited e.g. in table 3, O.R. for high LDL-C is mentioned as 1.01 with 95% C.I. of 1.00 to 1.01 and associated p-value <0.001 which is not possible and needs relooking. Also, none of the three tables given in this manuscript specifically deal with data by which MetS was calculated apart from calculating the five components of MetS as discussed before.

6. PLOS authors have the option to publish the peer review history of their article (what does this mean?). If published, this will include your full peer review and any attached files.

Reviewer #1: No

Reviewer #2: No

Reviewer #3: No

---

## [Author Response · Author response to Decision Letter 0]

13 Jun 2021

Dear Editor and Reviewer,

We are very grateful for the consideration of the manuscript. In accordance with the reviewers valuable comments and recommendations, we have revised the (MS 2) and we hereby re-submitting the revised work for your consideration.

Essential revision: Editor

English should be extensively revised.

-Introduction should be improved.

-Methodology needs important clarifications.

-Results and Discussion section should be more focused on the main results of the study.

Response: Comments are well taken we have obtained further editorial assistance to improve the language from native English spiker

Moreover, revision have been made in the abstract, Introduction, Methods result and discussion section

Response: The ORCID iD of the corresponding author is updated and validated as per the directions. 

Response: The questionnaire was directly adopted from the English version of the “WHO step by step approach for NCDs” and was translated to the local language called Amharic. We will also attach both the English and Amharic version if it helps as a supportive information.

4. The authors accept the comments and have removed the funding related information from the acknowledgment part. 

University of Gondar, a public higher education institute in Ethiopia, encourages the research culture of it’s academicians by arranging competitive partial grants for research projects. Less than 10% of research proposals get the partial grant (which does not include publication fee) and our research was granted through such small local grant. 

Therefore, the funding statement can be kept as it is as we have now omitted the funding information from the acknowledgement part. 

i.e. [No The funders had no role in study design, data collection and analysis, decision to publish, or preparation of the manuscript]

Response: done

The authors accept the comments and have removed the funding related information from the acknowledgment part. 

University of Gondar, a public higher education institute in Ethiopia, encourages the research culture of it’s academicians by arranging competitive partial grants for research projects. Less than 10% of research proposals get the partial grant (which does not include publication fee) and our research was granted through such small local grant. 

Therefore, the funding statement can be kept as it is as we have now omitted the funding information from the acknowledgement part. 

i.e. [No The funders had no role in study design, data collection and analysis, decision to publish, or preparation of the manuscript]

Reviewer 1

Reviewer #1: Thanks to the authors

It is better to correct and improve the English writing of the text.In the methodology section, the method of measuring the sample size is not completely clear

Please see the word's attached file.

Response: Comments are well noted, revision and correction are made. we have obtained further editorial assistance to improve the language

Reviewer 1 and 2: Thanks for inviting me to review this manuscript entitled "Magnitude of Metabolic Syndrome in Gondar Town, northwest Ethiopia: A community-based cross-sectional study". This manuscript reported a community-based cross-sectional study in Ethiopia (n=3059), which provided a picture for the prevalence of MetS in East Africa. Please see the attachment froe detailed comments.

Response: comments incorporation and revision done as per the comments. Please see the details below:

2. Careful editing is needed for the whole manuscript, such as the grammar, typos, e.g. Abstract-Conclusion and recommendation: line 6: “with high metabolic syndrome” ? The abbreviations should be explained with full names when they were firstly used, e.g. the IDF, NCD in the abstract.

Response: Comments are well taken, correction done in the revised section across the MS, The abbreviations are explained in full names in the abstract

1. Abstract: The study objectives were not presented clearly. “A community-based data on MetS” included what aspects? The results should be consistent with the objectives. 

Response: Done according to the suggestions. (comments are included in a track change)

2. Background: 

- The introduction should be presented more logistically. The rationales of conducting the current study should be clearly presented. From the current background, there were cross-sectional study, even population-based survey on MetS in Ethiopia. Why did the authors conducted the current one, using ATP and IDF definitions? 

Response: Comments are well taken additional points are incorporated to show the research gap (what is unknown in the background section) 

- P.10 of the submission file, 3rd paragraph, the sentence of “Moreover, to fully comprehend the standing of Dyslipidemia in Ethiopia population-centered epidemiological documented information is necessary and essential.” is abrupt.

Response: Correction done as suggested

- Many points in the introduction were not clear, such as the “rapid growth”, “inequitable distribution” “climatic changes”, “cultural variables”. 

Response: Comments are well taken those points are deleted for better clarity and focus

3. Methods and Material: 

- The time of conducting the study should be introduced. 

Response: 

Response: Well noted It’s 2018

- Sample size: why the author selected a “50% prevalence”? In the introduction, the prevalence ranged from 4.8%-20.3%. Sampling: how many people in each household were selected? “Finally, an individual from the household was selected using a random lottery method. All the individuals in each household were given an identification number by the data collector.” The two sentences were confusing. 

Response: The referred estimation sample size was taken in a pooled estimation prevalence, but ours sample size determination was done for each category of age and sex separately, and reported separately for those categories, Based on the suggestion revision done in the sapling process

- Measurements: The measurement methods for WC, body weight and height should be introduced. The references for categorizing these variables should be provided individually.

Response: Comments are incorporated in the revised MS measurement section

- Data management and analysis: the references for using “a cut of point p-value < 0.20” should be provided. 

Response: For the multivariable analysis to include the marginal confounder we use a wider cut of point < 0.20 (rule of thumb)

4. Results: 

- Why the authors used gender-and-age stratification in Table 1 and 2? This was not introduced previously?

Response: Study clearly indicate age and sex are predictors of MetS, in this study we want to show the distribution of MetS (prevalence of MetS across different age and sex group to have better information about the burden of the problem for public health intervention 

Some ideas are also introduced in the background section

- The whole data about MetS should be presented. For example, the mean, SD, and prevalence for BMI, WC, BP, FG, HDL, LDL, TG, and total cholesterol. The current manuscript only presented the categorical data (n, %). However, continuous data should be presented. 

Response: Comments are well taken additional table 4 is provided to address the comment

- Especially, as the IDF and ATP III used different WC criteria for abdominal obesity (94/80 or 102/88), prevalence of abdominal obesity following IDF and ATP III criteria should be provided. Otherwise, it is hard to understand the agreement of 91.7% between the two definitions.

Response: Prevalence of abdominal obesity following IDF and ATP III criteria, the proportion of Waist circumference using ATP III criteria was 12.7% [95%CI: 11.5, 13.9], and that of IDF was 26.9% [95%CI: 25.3, 28.5]. The agreement analysis/ computed was for MetS, but WC is one of the composite variable used to measure MetS;

- The risk factor for cardiovascular disease were not introduced previously. But Table 2 presented such data. 

- The regression model included variables with p<0.20. Where were the p values from? t-tests or ANOVA, or Chi square tests? Please clarify.

Response: The focus is not to discuss the risk factors for cardiovascular but just to show the whole picture of the burden 

Now we moved the table in to additional date if its not ok we can introduce detail in the main MS. The regression model we included variables with p<0.20 following univariate analysis in a Chi square tests.

- Multivariable analysis: One suggestion for the variable of marital status, the categories of separated, divorced, widowed could be combined. 

Response: Comments are well taken we recode to combine, the categories of separated, divorced, and widowed marital status and re run the regression (see table 3) 

- Table: there were typos in Table 1, e.g. Education status-Female: 214 (55,01); Marital status-Male : 29 (238). 

Response: Comments are well taken and correction done as suggested

- Figure: Figure 1 only presented the IDF criteria for age/gender prevalence of MetS, how about the ATP III criteria? Figure 2-4 presented the lipid files, how about other MetS components, like BP, WC, FG?

Response: We have incorporated additional figure (figure 5) to address the point

About other about other MetS components, like BP, WC, FG we have concern about the scope/focus we may end up with a lot of figure, we also show the mean in the new table 2

5. Discussion

- p.17 2nd paragraph: It is not appropriate to make direct comparison “a considerable rise” about MetS prevalence, considering the study variances in study time, the populations, criteria. 

Response: Revision is done based on the feedback

- It is also confusing that “the current study has included both adults and adolescents (> 17 years)” p.17; “a significant proportion of adolescents and young adults have metabolic syndrome” p.20. The study criteria were “aged 18 or older”.

Response: The current study is included only age 18 and above we do not have study participant from adolescents age 17 or less Correction is done for the word adolescents

- The discussion on “ frequent fasting was associated with high metabolic syndrome” was not clear. The possible reasons of this finding were not discussed. 

Response: Comments are well taken additional clarity point is included to address the comment

- The limitations in the current study were not discussed thoroughly.

Response: Done as commented (please the point in the limitation section

Reviewer #3

1. The introduction section does not amply describe a need to conduct a prevalence study on MetS on the said population. The authors themselves hinted on various such studies done already in this population and as per them, those seemed to have gaps in terms of lifestyle and anthropological perspectives; however, the authors in this study lacked to mention details on how those gaps were addressed in this study. 

Response: Comments are well taken, additional point and revision is done in the background section (please see the track change in the background)

2. The other concern regarding this study is the methodology used as the sampling design does not adequately cover the population due to the random lottery method used. The survey can also fail to cover the population as it does not categorize the people living in both standard and non-standard dwellings. This is an important factor for any community- based study.

Response: The study participants were resident of Gondar City, assuming a similar characteristics (homogenies) in terms of dietary practiced, and lifestyle;

For the sample size determination we used a separate sample size calculation/ determination considering the age and sex category, we also follow simple randomly sampling in order to select the representative study participant from each sub-city for age and sex group

3. Other issue about the methodology is the ATPIII criteria which is best predictor of the cardiovascular events but in elderly people. The conclusion drawn from the study thus cannot be extrapolated to other age-groups. Besides, the sample size calculation shown in the methodology section to select a representative sample for prevalence studies (especially when prevalence’s of >30% MetS have been reported in some studies) for a population of about 109 million needs revisiting.

Response: It is a well known that age by itself is an established predictor (non-modifiable risk factor) of cardiovascular event. ATPIII is one of the criteria used to diagnosed metabolic syndrome (various combinations of components of modifiable risk factors). We don’t is not a predictor by itself.

Having said that, we have estimation for each age category please see the newly add figure 5, generalization only for the respective category

Yes, conclusion drawn from the study cannot be extrapolated to other age-groups, the whole essence of this study is to show the distribution of MetS at different age group using the two criteria (ATPIII & IDF); the high prevalence is observed in older age group and we have seen association between the observed high prevalence with cardiovascular disease

4. The measurement section is a little confusing to the readers as the first sentence first sentence itself states that WHO and IDA criterion were used to classify hypertension while as per the abstract and rest of the manuscript, NCEP ATPIII or IDF were used to classify all the components of MetS. Also, the way of presenting the questionnaire details in the last paragraph of measurement section needs revisiting and should be presented in a more scientific manner.

Response: Correction done on WHO and IDF to avoid confusion, We use WHO step wise approach to collect data while letter we use ATP III and IDF criteria to classify MetS status (please see the revision in abstract and measurement section)

5. The result section should have been presented as and according to the data given in the tables and highlighting on the main findings which needed to be discussed in the discussion section. Also, the results should have been presented in accordance with the study objective which was to derive at the prevalence of MetS in the said population. MetS, according to the criteria used, has components like “central obesity”, “hypertension”, “hypertriglyceridemia”, “low HDL-cholesterol” and “hyperglycemia” and thus focus should have been on these components and the full MetS rather than describing hypercholesteromia, high LDL-C, etc. 

Response: Comments are well taken and revision done focusing on the key findings, and its order (please see the rearrangement in the revised version with a track change)

6. The discussion lacks the details of the defense of the observations and findings in the current study. For example, the initial statements in the discussion relates the findings in the current study in line with a study (reference # 20) while the sample size in that study was 314 and the prevalence of MetS observed was 59.9% and 70.1% according to IDF and NCEP criteria’s respectively. Besides, the authors struggled in explaining the reason for such a low prevalence of MetS (as we have numerous reports of much higher prevalence of MetS in the general population worldwide).

Response: Considering your valuable comment we have tried to incorporate findings from Ethiopia to show context relation or difference with possible explanation and its implication (please see the change in the second paragraph of the discussion section)

7. Lastly, there are numerous grammatical mistakes in the entire manuscript which should be looked at by the authors seriously before resubmission. Besides, the statistics done should be revisited e.g. in table 3, O.R. for high LDL-C is mentioned as 1.01 with 95% C.I. of 1.00 to 1.01 and associated p-value <0.001 which is not possible and needs relooking. Also, none of the three tables given in this manuscript specifically deal with data by which MetS was calculated apart from calculating the five components of MetS as discussed before.

Response: Comments are well taken, correction done in addition figure 5 is incorporated to address the issue (Please see figures 5)

With Best Regard!! 

The Authors

---

## [Decision Letter · Decision Letter 1]

25 Jun 2021

PONE-D-21-02096R1

Magnitude of Metabolic Syndrome in Gondar Town, northwest Ethiopia: A community-based cross-sectional study

PLOS ONE

Dear Dr. Solomon Mekonnen Abebe,

Thank you for submitting your manuscript to PLOS ONE. After careful consideration, we feel that it has merit but does not fully meet PLOS ONE’s publication criteria as it currently stands. Therefore, we invite you to submit a revised version of the manuscript that addresses the points raised during the review process.

This editor as well as the reviewer feel that not all essential previopus suggestions have been considered, or a proper reply has not been provided. English still requires a massive improvement, and sections such as introdution, methods and results should be clearly improves followin the reviewers' comments.

We look forward to receiving your revised manuscript.

Kind regards,

Pedro Tauler, Ph.D.

Academic Editor

PLOS ONE

Reviewers' comments:

Reviewer's Responses to Questions

**Comments to the Author**

1. If the authors have adequately addressed your comments raised in a previous round of review and you feel that this manuscript is now acceptable for publication, you may indicate that here to bypass the “Comments to the Author” section, enter your conflict of interest statement in the “Confidential to Editor” section, and submit your "Accept" recommendation.

Reviewer #1: All comments have been addressed

Reviewer #2: (No Response)

Reviewer #3: All comments have been addressed

2. Is the manuscript technically sound, and do the data support the conclusions?

Reviewer #1: No

Reviewer #2: Yes

Reviewer #3: Yes

3. Has the statistical analysis been performed appropriately and rigorously? 

Reviewer #1: No

Reviewer #2: Yes

Reviewer #3: Yes

4. Have the authors made all data underlying the findings in their manuscript fully available?

Reviewer #1: No

Reviewer #2: Yes

Reviewer #3: Yes

5. Is the manuscript presented in an intelligible fashion and written in standard English?

Reviewer #1: No

Reviewer #2: No

Reviewer #3: Yes

6. Review Comments to the Author

Reviewer #1: Unfortunately, despite the renewed opportunity given to the authors, it still seems that the criteria of the journal have not been met. I hope to see his next works in the near future

Reviewer #2: Thanks for inviting me to review the revised manuscript. The authors made revisions following the previous comments. However, there are some comments that were not fully responded.

1. Careful editing is still needed for the whole manuscript, such as the grammar, punctuation, reference citation, repeated numbers. Please pay attention to the used of abbreviations in the main text.

2.Background:

- please explain some points in the introduction, such as the“climatic changes”, “dietary practice”.

3.Methods and Material:

-Sample size: why the author selected a “50% prevalence”? In the introduction, the prevalence ranged from 4.8%-20.3%. although the authors explained the age-stratification in sample size calculation, which age-group assumed the 50% prevalence of MetS (providing the reference)? Please also introduce the six age-groups.

- Measurement: “Finally, biochemical tests (Impaired fasting blood glucose levels (IFG) were between 110 and 125 mg/dl....) were carried out.” is confusing.

- Data management and analysis: the references for using “a cut of point p-value < 0.20” should be provided.

4.Results:

- The data in Table 1 were not consistent. E.g. the “location birth” variable showed a total of 3062 participants, with 1406 males and 1656 females. But the “Education status” variable showed a total of 2977 participants, with 1388 males and 1589 females. Moreover, 214 (55,0) still exist. The total numbers were not 3059, as the authors introduced. Please check all the data.

-Table 2, the units of the variables were not presented.

- please check the sentence in the last paragraph of Results Session: “Similarly, A unit increase in Body mass index of body weight per cm2 will increase Metabolic Syndrome by 18%

[AOR= 18%; 95%CI:1.15, 1.22]. ”

-When reporting the multivariable analysis findings, please clarify which MetS definition was used.

5.Conclusion

-“Though less prevalent than the general population, a significant proportion of adolescents and young adults have metabolic syndrome.” This was not appropriate, as no adolescents were recruited in the study participants.

Reviewer #3: 1. The refrences in the background section doesnot reflect the same cohort nor age or trype of study identical epidemonological studies capturing equvivalent age group 18 years and above and reflecting other ethinicities is what is needed for comaprision.

2. Rephrase some words like 'being targeted by marketing' mentioned in the next paragraph.

3. Randomly house selection used in the study method has to be clearly mentioned and rephrasing the methadology section accordingly.

4. In the result section occupation should replace main work type.

general comments include the following:

A. English language has to be improved.

B. Typo error has to be corrected

C. Using metabolic syndrome word is sometime written as MetS,author has to fix it ,preferably using full name.

7. PLOS authors have the option to publish the peer review history of their article (what does this mean?). If published, this will include your full peer review and any attached files.

Reviewer #1: No

Reviewer #2: No

Reviewer #3: No

---

## [Author Response · Author response to Decision Letter 1]

20 Jul 2021

To: Plos one Editor-in-Chief 

Subject: -Point by point response for the comments and recommendations of

reviewers 

Title: 

Reference: MS: 

Dear Editor-in-Chief

We are very grateful for the consideration of the manuscript. In accordance with the reviewers valuable comments and recommendations, we have revised the (MS3) and we hereby re-submitting the revised work for your consideration.

PONE-D-21-02096R1

Magnitude of Metabolic Syndrome in Gondar Town, northwest Ethiopia: A community-based cross-sectional study

PLOS ONE

Essential revision

Editor 

1. Review Comments to the Author

Response: Comments are well taken, this MS is not published elsewhere or submitted to other Journal; The University of Gondar IRB provide ethical clearance and aware of the MS submission for publication 

Reviewer 2

2. Careful editing is still needed for the whole manuscript, such as the grammar, punctuation, reference citation, repeated numbers. Please pay attention to the used of abbreviations in the main text.

Response: The authors accept the comments and have obtained further editorial assistance to improve the language. 

2.Background:

- please explain some points in the introduction, such as the“climatic changes”, “dietary practice”.

Response: Climatic changes and dietary practice mentioned in the introduction part as per the comment.

3.Methods and Material:

-Sample size: why the author selected a “50% prevalence”? In the introduction, the prevalence ranged from 4.8%-20.3%. although the authors explained the age-stratification in sample size calculation, which age-group assumed the 50% prevalence of MetS (providing the reference)? Please also introduce the six age-groups.

Response: Yes, there is a research having estimated prevalence ranged from 4.8%-20.3%; however, it is a pooled estimation for all age and sex, it does not account the above variables (without stratification). But our sample size determination used a separate sample size calculation/ determination considering the age and sex category, 

(Please refer WHO NCDs sample size assumption which most literature used with large sample size (World Health Organization, author. Chronic diseases and health promotion STEPwise approach to surveillance (STEPS) STEPS Manual. [December 2005]. Available at: http://www..who.int/chp/steps. [Google Scholar]

3.2. Measurement: “Finally, biochemical tests (Impaired fasting blood glucose levels (IFG) were between 110 and 125 mg/dl....) were carried out.” is confusing.

Response: comments well taken revision done as per the comments, for farther clarity in the revised version,

3.3. - Data management and analysis: the references for using “a cut of point p-value <cdct” should be provided.

Response: It’s a rule of tube used by many research as a cut of value to account marginal confounder that may confound the independent variables with the outcome we use inter method while we fit the regression model in the case of backward and forward the approach is totally different

4.Results:

4.1 - The data in Table 1 were not consistent. E.g. the “location birth” variable showed a total of 3062 participants, with 1406 males and 1656 females. But the “Education status” variable showed a total of 2977 participants, with 1388 males and 1589 females. Moreover, 214 (55,0) still exist. The total numbers were not 3059, as the authors introduced. Please check all the data. 

Response: Thank you for the comment, correction done (please see the track change in the Table 

4.2 Table 2, the units of the variables were not presented.

Response: Comments are well taken; Units are included in revised Table 2

4.3 Please check the sentence in the last paragraph of Results Session: “Similarly, A unit increase in Body mass index of body weight per cm2 will increase Metabolic Syndrome by 18%

[AOR= 18%; 95%CI:1.15, 1.22]. ” -When reporting the multivariable analysis findings, please clarify which MetS definition was used. 

Response: We use IDF classification for the regression report, comments are well taken and correction is done

5.Conclusion

-“Though less prevalent than the general population, a significant proportion of adolescents and young adults have metabolic syndrome.” This was not appropriate, as no adolescents were recruited in the study participants.

Response: Yes, (we do not have adolescents) Correction is done as per the comment

Reviewer #3: 

1. The refrences in the background section doesnot reflect the same cohort nor age or trype of study identical epidemonological studies capturing equvivalent age group 18 years and above and reflecting other ethinicities is what is needed for comaprision.

Response: Unfortunately, the previous studies didn’t have as much age spectrum unlike ours; and that was one of the justifications for the uniqueness of this study in including a larger age spectrum.

2. Rephrase some words like 'being targeted by marketing' mentioned in the next paragraph.

Response: Considering your valuable comment we have tried to correct the wording

3. Randomly house selection used in the study method has to be clearly mentioned and rephrasing the methadology section accordingly.

Response: Comments are well taken, correction done

4. 4. In the result section occupation should replace main work type.

general comments include the following:

5. Response: Correction done as suggested

6. A. English language has to be improved.

Response: Comments are well taken, additional copyedit is done to improve the language

B. Typo error has to be corrected 

Response: Done

C. Using metabolic syndrome word is sometime written as MetS,author has to fix it ,preferably using full name

Response: Comments are well taken MetS are replaced by metabolic syndrome in the main document (except the abstract) 

-The discussion on “frequent fasting was associated with high metabolic syndrome” was not clear. The possible reasons of this finding were not discussed. 

Response: It seems a narration error. The multivariate analysis (Table 3) actually shows that not fasting is associated with metabolic syndrome compared to those who fast. Correction is made on the discussion part based according to the findings on Table 3 based on the comments.

- The limitations in the current study were not discussed thoroughly.

Response: Adding more limitations to the current study was tried on the discussion part as per the comment.

 With Best Regard!!

 The Authors

---

## [Decision Letter · Decision Letter 2]

7 Aug 2021

PONE-D-21-02096R2

Magnitude of Metabolic Syndrome in Gondar Town, northwest Ethiopia: A community-based cross-sectional study

PLOS ONE

Dear Dr.Solomon Mekonnen Abebe,

Thank you for submitting your manuscript to PLOS ONE. After careful consideration, we feel that it has merit but does not fully meet PLOS ONE’s publication criteria as it currently stands. Therefore, we invite you to submit a revised version of the manuscript that addresses the minor editing points raised during the review process by reviewer 2.

We look forward to receiving your revised manuscript.

Kind regards,

Pedro Tauler, Ph.D.

Academic Editor

PLOS ONE

Journal Requirements:

Reviewers' comments:

Reviewer's Responses to Questions

**Comments to the Author**

1. If the authors have adequately addressed your comments raised in a previous round of review and you feel that this manuscript is now acceptable for publication, you may indicate that here to bypass the “Comments to the Author” section, enter your conflict of interest statement in the “Confidential to Editor” section, and submit your "Accept" recommendation.

Reviewer #1: All comments have been addressed

Reviewer #2: All comments have been addressed

Reviewer #3: All comments have been addressed

2. Is the manuscript technically sound, and do the data support the conclusions?

Reviewer #1: Yes

Reviewer #2: Partly

Reviewer #3: Yes

3. Has the statistical analysis been performed appropriately and rigorously? 

Reviewer #1: I Don't Know

Reviewer #2: Yes

Reviewer #3: Yes

4. Have the authors made all data underlying the findings in their manuscript fully available?

Reviewer #1: Yes

Reviewer #2: Yes

Reviewer #3: Yes

5. Is the manuscript presented in an intelligible fashion and written in standard English?

Reviewer #1: Yes

Reviewer #2: No

Reviewer #3: Yes

6. Review Comments to the Author

Reviewer #1: Nothing Thank you for your attention to correct process reviewing. please keep safety and health in this cituation.

Reviewer #2: The authors made some revisions following the previous comments. However, there are still some questions needs clarification and improvements. The authors should point out the detailed location of each revision in the responses (e.g. page no. Paragraph, line).

1. Careful editing is still needed for the whole manuscript. Please pay attention to the following:

--It should be “metabolic syndrome” but not “Metabolic syndrome” if this word is not in the beginning of the sentence.

-P.41 the last 2nd line, it should be “such as obesity...” .

- Measurement: “Finally, biochemical tests (fasting blood glucose levels....) were carried out. Impaired fasting blood glucose levels (IFG) was identified if the fasting blood glucose levels were between 110 and 125 mg/dl.

- Results Session: “A unit increase in LDLC/ BMI...will increase the prevalence of metabolic Syndrome by XX%. ”

-Table 2: “Standard deviation”

- Discussion: P.50, line 2 : “ a higher prevalence”

2.Sample size: the rationales or references of using a “50% prevalence” for the age-stratified sample size calculation should be provided.

Reviewer #3: The paper is generally well written and structured. The changes are well addressed by author, thus I recommend the acceptance of the article.

7. PLOS authors have the option to publish the peer review history of their article (what does this mean?). If published, this will include your full peer review and any attached files.

Reviewer #1: No

Reviewer #2: No

Reviewer #3: No

---

## [Author Response · Author response to Decision Letter 2]

27 Aug 2021

Dear Reviewer,

We are very grateful for the consideration of the manuscript. In accordance with the reviewer valuable comments and recommendations, we have revised the (MS 4) and we hereby re-submitting the revised work for your consideration.

PONE-D-21-02096R1

Magnitude of Metabolic Syndrome in Gondar Town, northwest Ethiopia: A community-based cross-sectional study

PLOS ONE

Editor:

Thanks for inviting me to review the revised manuscript. The authors made some revisions following the previous comments. However, there are still some questions needs clarification and improvements. The authors should point out the detailed location of each revision in the responses (e.g. page no. Paragraph, line).

 Careful editing is still needed for the whole manuscript. Please pay attention to the following:

Response: The authors accept the comments and have obtained further edition to improve the language.

Reviewer comment

1. It should be “metabolic syndrome” but not “Metabolic syndrome” if this word is not in the beginning of the sentence.

Response: Comments well taken revision done as per the comments 

2. P.41 the last 2nd line, it should be “such as obesity...” .

Response: Thank you for the comment, correction done (please see the track change in the revised version)

3. Measurement: “Finally, biochemical tests (fasting blood glucose levels....) were carried out. Impaired fasting blood glucose levels (IFG) was identified if the fasting blood glucose levels were between 110 and 125 mg/dl.

Response: Thank you for the comment, correction done (please see the track change in the revised version)

4. Results Session: “A unit increase in LDLC/ BMI...will increase the prevalence of metabolic Syndrome by XX%. ”

Response: Comments are well taken; correction done ( the prevalence)

5. Table 2: “Standard deviation”

Response: Thank you for the comment, spelling correction done (standard deviation)

6. Discussion: P.50, line 2 : “ a higher prevalence”

Response: Thank you for the comment, spelling correction done Higher

7. Sample size: the rationales or references of using a “50% prevalence” for the age-stratified sample size calculation should be provided

Response: Comments are well taken, the reference is cited : World Health Organization. Noncommunicable Diseases and Mental Health Cluster. (‎2005)‎. WHO STEPS surveillance manual : the WHO STEPwise approach to chronic disease risk factor surveillance / Noncommunicable Diseases and Mental Health, World Health Organization. World Health Organization. https://apps.who.int/iris/handle/10665/43376,

With Best Regard!!

The Authors

---

## [Editor Report · Decision Letter 3]

31 Aug 2021

Magnitude of Metabolic Syndrome in Gondar Town, northwest Ethiopia: A community-based cross-sectional study

PONE-D-21-02096R3

Dear Dr. Solomon Mekonnen Abebe,

We’re pleased to inform you that your manuscript has been judged scientifically suitable for publication and will be formally accepted for publication once it meets all outstanding technical requirements.

Kind regards,

Pedro Tauler, Ph.D.

Academic Editor

PLOS ONE
---

## [Editor Report · Acceptance letter]

20 Sep 2021

PONE-D-21-02096R3 

Magnitude of Metabolic Syndrome in Gondar Town, Northwest Ethiopia: A Community-based Cross-sectional Study 

Dear Dr. Abebe:

I'm pleased to inform you that your manuscript has been deemed suitable for publication in PLOS ONE. Congratulations! Your manuscript is now with our production department. 

Kind regards, 

on behalf of

Dr. Pedro Tauler 

Academic Editor

PLOS ONE